REGISTERED REPORT PROTOCOL

# The expectations and acceptability of a smart nursing home model among Chinese elderly people: A mixed methods study protocol

Yuanyuan Zhao[1,2], Shariff-Ghazali Sazlina[1,3], Fakhrul Zaman Rokhani[3,4], Jing Su[5], Boon-How Chew[1,6]*

1 Faculty of Medicine and Health Sciences, Department of Family Medicine, Universiti Putra Malaysia, Serdang, Selangor, Malaysia, 2 Global Century Science Group, Shenyang, China, 3 Malaysian Research Institute on Ageing, Universiti Putra Malaysia, Persiaran MARDI - UPM, Serdang, Selangor, Malaysia, 4 Faculty of Engineering, Department of Computer and Communication Systems Engineering, Universiti Putra Malaysia, Serdang, Selangor, Malaysia, 5 College of Public Health, Hainan Medical University, Haikou, China, 6 Clinical Research Unit, Hospital Pengajar Universiti Putra Malaysia (HPUPM Teaching Hospital), Persiaran MARDI - UPM, Serdang, Selangor, Malaysia

* chewboonhow@upm.edu.my

## Abstract

Nursing homes integrated with smart information such as the Internet of Things, cloud computing, artificial intelligence, and digital health could improve not only the quality of care but also benefit the residents and health professionals by providing effective care and efficient medical services. However, a clear concept of a smart nursing home, the expectations and acceptability from the perspectives of the elderly people and their family members are still unclear. In addition, instruments to measure the expectations and acceptability of a smart nursing home are also lacking. The study aims to explore and determine the levels of these expectations, acceptability and the associated sociodemographic factors. This exploratory sequential mixed methods study comprises a qualitative study which will be conducted through a semi-structured interview to explore the expectations and acceptability of a smart nursing home among Chinese elderly people and their family members (Phase I). Next, a questionnaire will be developed and validated based on the results of a qualitative study in Phase I and a preceding scoping review on smart nursing homes by the same authors (Phase II). Lastly, a nationwide survey will be carried out to examine the levels of expectations and acceptability, and the associated sociodemographic factors with the different categories of expectations and acceptability (Phase III). With a better understanding of the Chinese elderly people's expectations and acceptability of smart technologies in nursing homes, a feasible smart nursing home model that incorporates appropriate technologies, integrates needed medical services and business concepts could be formulated and tested as a solution for the rapidly ageing societies in many developed and developing countries.

**Data Availability Statement:** All relevant data from this study will be made available upon study completion.

**Funding:** The authors received no specific funding for this work.

**Competing interests:** The authors have declared that no competing interests exist.

## Introduction

The oldest nursing homes known as 'gerocomeia' in Greek, were constructed in the 5th century in Byzantium to care for the elderly people in special infirmaries [1]. Nowadays, 'nursing home' is a concept defined as a facility with a domestic-style environment that provides 24-hour functional care for elderly people who require assistance with daily living activities and who often have complex health needs [2]. Thus, today's nursing homes also provide some degree of medical services from healthcare professionals. Residency in a nursing home may be relatively brief for respite purposes or long term, particularly for those requiring palliative/hospice and end-of-life care [2]. Admission to a nursing home is usually a major life event for most elderly people due to the several triggered events such as retirement, departure of children from home, death of a spouse or loss of physical ability [3,4].

Several countries across the world, including China, are facing complex challenges of rapid ageing [5]. The National Bureau of Statistics of China (2018) revealed that the number of elderly people aged ≥65 years old had reached 167 million (12%) in 2018, and China will become a super-aged society with more than 21% elderly population by 2035 [6]. The ageing population in France, in contrast, took 100 years to progress from 7% to 14% of the total population [7]. Influenced by the Confucian philosophy of filial piety which teaches the virtue of 'Xiao' and respect for one's parents and the elders [8], most elderly Asians expect 'ageing in place' [9]. Nevertheless, it has become hardly feasible for a single child or grandchild to provide care for two parents or four grandparents in China's one-child policy [10]. In 2019, up to 76% of the Chinese elderly people aged ≥60 years old had at least one chronic disease such as cardiovascular diseases, obstructive respiratory diseases or type 2 diabetes [11], and the population aged ≥80 years old has doubled the number of those aged ≥65 years old, and their medical demand is five times higher than those aged between 65 and 79 years old [12]. More elderly people in China are accepting nursing homes as an alternative to ageing in their homes due to the consequences of the one-child policy and the high cost of hiring home carers [10]. In 2015, the Chinese government reported that the capacity of nursing beds at nursing homes was about 5.5 million, reflecting an average of 26 beds per 1000 elderly people [13]. Besides, the quality of care and medical services offered by health professionals in most nursing homes is largely suboptimal [14]. Hence, many elderly residents in nursing homes still need to visit the hospitals to fulfil their healthcare needs [14,15].

There are many factors that influence the consideration for choosing nursing homes. Most elderly people and their family members prefer urban nursing homes as they offer better quality healthcare services provided by professional nurses and medical practitioners, or simply because they are located nearby hospitals that provide integrated medical services and first aid treatment for medical emergencies [16]. The elderly people also consider a nursing home that is easily accessible and that could offer an alternative living environment when they experience a declining health condition [15]. However, fears and negative perceptions about nursing homes in terms of the quality of care, quality of life and loss of relationships with family members are real among the elderly people [17]. In addition, affordability of a nursing home and medical services are the other two main concerns among the Chinese elderly people when considering about living in a nursing home [18]. The average monthly cost of a typical nursing home in China is between RMB 1000 and RMB 4000 ($142-$570) per month, and the national insurance does not cover this long-term healthcare [19].

To overcome these personal and social challenges, the Chinese government initiated 'The Healthy China 2030' and 'Integrating Medical Care and Senior Services' as the guiding principles in addressing the integration of medical resources into the healthcare sector in order to

better serve the ageing population [20]. Evidence has shown that 'smart' information technologies could utilize medical resources to their maximal potential without geographical barriers and assist elderly patients in leasing an active, fulfilling, and good quality life [21,22]. Smart healthcare which utilizes the new generation of information technologies such as the Internet of Things (IoT), big data, cloud computing, and artificial intelligence could transform the conventional medical healthcare, making it more efficient, convenient, and personalised [23]. The World Health Organization (WHO) (2019) associates smart technologies in healthcare with digital health including telemedicine and mobile applications for healthcare [24]. Smart technologies encourage 'ageing in place' [25] and the use of a variety of technologies such as the different types of active and passive sensors, monitoring devices, and the use of robotics in a home setting [26]. In 2015, the Chinese government proposed the 'Internet Plus' plan to address technology innovation and initiate the integration of social services in building of smart homes for elderly care [27]. The 'Law of the People's Republic of China on the Protection of the Rights and Interests of the Elderly' was amended in 2018 to emphasize the principles for home-based elderly care so that they rely more on the community and at the same time to encourage institutions to develop new nursing home models to provide care for the ageing population [28]. 'Ageing in place' is desirable; however, many elderly people may require a transition from home-based care to institutional care with a 24-hour supervision due to changes in health status or non-availability of family members to care for them at home [29]. However, the existing nursing homes in China are policy-driven rather than demand-driven, and they are not technologically smart or meet the expectations of the health conscious elderly people and their family members. This has brought about many challenges to a balanced development, quality maintenance, and sustainability of the 'traditional' nursing homes [30].

Nursing homes that integrate smart information and medical technologies appropriately may benefit the residents and nursing staffs as well as health professionals by providing effective care and efficient medical services [31–33]. The adoption of smart technologies in nursing homes has not only improved the management of chronic diseases, but also made it more cost-effective [31,34,35]. These technologies include telemedicine [36–38], mHealth applications [31,39], and wearable [40,41] and monitoring devices through sensors [42–44]. The IoT, wearable technology, and mobile technology could be used to provide real-time, efficient, and intelligent monitoring services in the nursing homes [45]. However, few studies have formulated the framework or architecture of a smart nursing home based on IoT [46–48], and fewer have integrated medical services or hospitals to meet the diverse needs of the elderly people with chronic health conditions [49,50]. In 2014, the Ministry of Civil Affairs, China released the 'Smart Elderly Internet of Things Pilot Project' to encourage the development of smart nursing homes [51]. This policy emphasized that any projects based on the IoT and artificial intelligence (AI), or related to posture monitoring and location systems, fall protection, activity analysis, automatic diagnosis of diseases, and video integration for elderly care would be supported as pilot projects of smart nursing homes.

Previous studies conducted on the topics [40,52–54] focused on the smart solutions for nursing home residents, expectations of the services, and users' acceptance of the technologies. However, the perception of a nursing home that is highly connected and equipped with advanced information technologies related to personal monitoring and medical services among the mainland Chinese elderly people is lacking in the literature. Therefore, the expectations and acceptability of smart nursing homes from the perspectives of stakeholders especially the target users, for example, the elderly people and their family members, are imperative for the development of a successful smart nursing home model.

Expectations  Adoption Processes  Outcomes

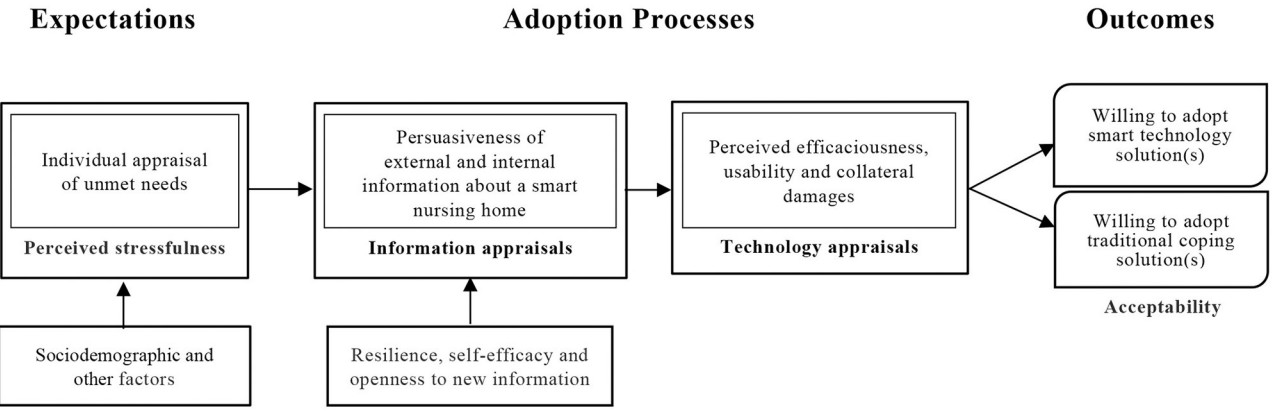

**Fig 1. The theoretical model of smart technology adoption behaviours by the elderly consumers.**

## Theoretical framework

The theoretical model by Golant (2017) on smart technology adoption behaviours by the elderly consumers will be used to guide the analysis and interpretation of data (Fig 1) [55]. Based on the technology acceptance model (TAM) [56], the unified theory of acceptance (UTAUT) [57], the use of technology model (UTAUT2) [58] and their theoretical underpinnings [59], including stress-appraisal coping theory [60], the health belief model [61], diffusion theory [62], information-seeking models [63,64], personality differences [65], life span coping process dynamics [66], and, social cognitive theories, particularly persuasion and social influence models [67–69], this theoretical model drew on the constructs and relationships from a wide array of marketing, communication, and social psychological theories. The model proposed by Golant [55] included the evaluation of individual effects and environmental influences on the coping process of the technology, and it is also applicable in the analysis and interpretation of the perspectives of family members during the appraisal of the technologies to be used in the care of the elderly people. As the theoretical model suggests that elderly people will be more acceptable to smart technologies if they have unmet ageing needs and would feel stressed in finding alternative solutions. Those with greater resilience would have stronger perceptions of self-efficacy and greater openness to new information; thus, they will be more likely to evaluate the external information of the technologies. The coping process refers to how family members and health professionals react to the technologies and smart nursing homes (persuasiveness of external information), in particular with regard to the perceived efficaciousness, usability, and collateral damages until the elderly people reach the stage of coming to a decision, for example, whether to adopt the 'new' smart technologies or to maintain the 'old' ways of life. This behavioural process will be influenced by their past experiences (internal information). As for the persuasiveness of the external information to be possible, a clear concept of a smart nursing home is necessary. Technology appraisals are feasible when the range of technologies that have been proven to be applicable and safe for use in nursing homes. Expectations and acceptability of nursing home stakeholders especially the elderly people, will also be affected by other factors such as sociodemographic characteristics, perceived quality of care, cost, and general ambience and vicinity of the nursing homes to the family members [70]. To be specific, expectations are the desires or wants of consumers of what they feel a smart nursing home should offer [71] while acceptability is a priori phenomenon of adoption and refers to intention to use or willingness to adopt smart nursing homes [72,73]. Thus, expectations and acceptability are not two separate constructs; expectation is a determinant

of acceptability if the smart nursing home appraisal meets the expectations [74]. The study hypothesizes that having an expectation of a smart nursing home is associated with better resilience to external and internal changes in need appraisal and increases the acceptability of the smart nursing homes and intention of adopt smart solutions for elderly care. Hence, this model is appropriate in guiding the formulation of the research objectives. The theoretical model will be explored through this study and adapted as necessary based on the findings.

### Research questions

The general research question in this study is: Could a smart nursing home model be technologically feasible, integrated with medical services and accepted by all stakeholders in China? A preceding scoping review [75] will provide a clear definition and a general concept of smart nursing homes designed for relatively well elderly people without terminal illness, not bed-bound or requiring critical care. These nursing homes are not ICU/HDU units. The range of technological approaches that make smart nursing homes 'smart' include the following quality healthcare elements: safe, effective, efficient, equitable and people-centred [76]. The following specific review questions are formulated: 1. What are the expectations and demands of a smart nursing home among the Chinese elderly people and their family members?; 2. Will they accept and how will they accept the evidence-based smart nursing home model?; and 3. What is the association between the sociodemographic characteristics of Chinese elderly people and their expectations and acceptability? Guided by the theoretical model by Golant (2017) on smart technology adoption behaviours by the elderly consumers, this study aims to explore the expectations and acceptability of a smart nursing home from the perspectives of mainland Chinese elderly people, and to determine the levels of these expectations and acceptability, and the associated sociodemographic factors.

### Study objectives

1. To explore the expectations of a smart nursing home among the mainland Chinese elderly people and their family members.

2. To explore the acceptability of a smart nursing home by stakeholders, especially the elderly people and their family members.

3. To develop a questionnaire based on the scoping review and a qualitative study to assess the expectations and acceptability of nursing homes in general and a smart nursing home in particular.

4. To examine the levels of expectations and acceptability of a smart nursing home among the mainland Chinese elderly people through a nationwide online questionnaire.

5. To determine the sociodemographic factors associated with the different categories of expectations of a smart nursing home.

6. To determine the sociodemographic factors associated with the different categories of acceptability of a smart nursing home.

### Methods

Based on the research objectives, this exploratory sequential mixed methods study will consist of three phases: qualitative study (Phase I), questionnaire development and its validation (Phase II), and quantitative study (Phase III). Through the semi-structured interviews in Phase

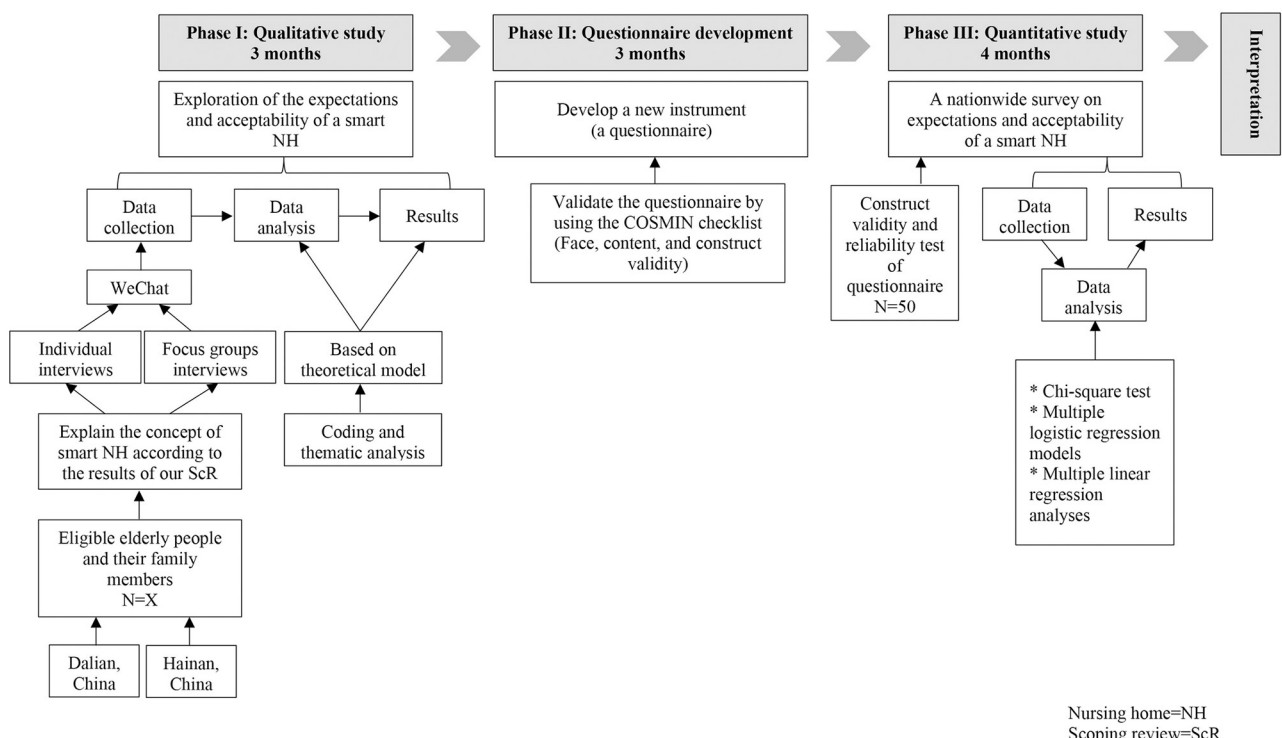

**Fig 2. Exploratory sequential mixed methods study design.**

I, we will explore the variables of expectations and acceptability of a smart nursing home among the purposive samples of the Chinese elderly people. In Phase II, a questionnaire will be developed and validated as the instrument in Phase III. In Phase III, the study will examine the levels of expectations and acceptability, and the sociodemographic factors associated with the different categories of expectations and acceptability of a smart nursing home through a nationwide survey. Fig 2 demonstrates the exploratory sequential mixed methods study design.

Ethical approvals for this study have been obtained from the Ethics Committee for Research Involving Human Subjects, Universiti Putra Malaysia, Malaysia (UPM/TNCPI/RMC/JKEUPM/1.4.18.2, 28/11/2020) and Hainan Medical University, China (IYLIJ-2020-021, 03/09/2020). The Respondent's Information Sheet will be provided to the participants and an Informed Consent Form will be required before participation.

## Phase I: Exploration of the expectations and acceptability of a smart nursing home from the perspectives of the mainland Chinese elderly people

In Phase I, a semi-structured interview involving mainland Chinese elderly people and the family members of eligible elderly people will be conducted to address research objectives 1 and 2. The qualitative study will provide insights on the needs of the Chinese elderly people, their expectations of a smart nursing home, and the acceptability toward smart technologies to improve the quality of care and integration of medical services. After undergoing training in conducting qualitative studies, two investigators (ZYY and SJ) will conduct interviews in the field.

**Setting.**   In this research, eligible participants in selected sites: Hainan Island and Dalian city in mainland China will be recruited. Hainan Island is a province situated in the South of mainland China. Its tropical climate with an average temperature above 22°C in winter endows the island with a unique natural environment. Every October, there are 1.64 million 'migratory birds' of elderly people (17% of the population in Hainan) from 27 Chinese provinces come to Hainan Island for winter and only return home in April the following year [77,78]. These elderly people commented that their chronic diseases would be significantly alleviated when they spend winter in Hainan Island [79]. Dalian is chosen as the second site for data collection as it is located in Northeast China, and it has a high proportion of ageing population [80]. The municipal government of Dalian has high regards for senior care and nursing home projects to ensure sufficient bed capacity in the city [81,82]. It is one of the beautiful cities in China and is an internationally renowned eco-liveable city [83]. The weather in Dalian is pleasant. It has four distinct seasons. The summer is not too hot, and the winter is not too cold [84]. It is also known as one of the 'top 10 liveable cities' in China for elderly people to settle down after retirement [85]. Dalian attracts rich elderly people from other provinces to either buy their own apartments or reside in a nursing home in Dalian [86].

**Participants.**   The inclusion criteria for recruiting the purposive samples will be: 1) migrants aged 60–75 years old from other parts of China (citizens) living in Hainan Island and Dalian; 2) local residents living in Hainan and Dalian; and 3) those with or without chronic diseases; 4) those having only one child or two children who may have difficulty in providing care for their parents. To be excluded are elderly people who are not citizens of China and those who are unwilling to be interviewed or are not able to communicate clearly due to language problems, such as dysphagia and dementia. In order to get more insights on the expectations and acceptability of smart nursing homes, the family members of the eligible elderly people will also be included in the interview. The inclusion criteria include: 1) being employed or self-employed; 2) aged ≥30 years old; and 3) having relatives who stay in the same house with their parents or take care of their parents by helping them with certain chores such as cooking and cleaning the house, or assisting their parents in daily routines. Other family members of eligible elderly people including grandchildren and children-in-law will be excluded.

**Sampling process.**   The time frame for Phase I is estimated to last for 3 months. We will conduct the snowball sampling method to recruit the samples [87]. The investigators (ZYY and SJ) will recruit participants in Hainan Island and Dalian based on the inclusion and exclusion criteria. The interviewees will be asked to recommend other eligible participants among their friends and neighbours who wish to participate in this research. One investigator (ZYY) will recruit eligible samples from Dongfang, Hainan Island, and Dalian. The co-investigator (SJ) will help to select eligible samples from Haikou, Hainan Island. Meanwhile, research information will be distributed in the WeChat groups of the elderly communities. WeChat is a common communication App in China. Chat groups are created by friends, residents, and the communities, and each WeChat group may consist of a maximum of five hundred members. WeChat has similar functions as Facebook and WhatsApp. Investigators will contact eligible participants to explain the study details, confirm their interest in participating, and determine their preferred interview method, for example, individual interview, focus group discussion, voice call or voice messages on WeChat. Payment to participants will be transferred online via WeChat wallet (the most common electronic payment method in China).

**Data collection.**   Data collection in Phase I will involve two types of data collection methods: in-depth interviews and focus groups using semi-structured interview guides. Further, in-depth interviews may be conducted with certain respondents who have attended the previous focus group interviews in order to gain more perspectives and to clarify the findings from the

focus groups. Each focus group will consist of 4–6 participants. An earlier scoping review conducted by the same research team will help to define the concept and criteria of a smart nursing home. The findings will provide additional information in developing the interview guides and questions; for example, 'Can you talk about your perception of a smart nursing home?', 'Do you think a nursing home should have integrated medical services?', 'What do you expect from a smart nursing home?', 'Would you allow yourself to be admitted to a smart nursing home? If Yes, Why? If not, Why?' The preliminary semi-structured interview guides (shown in S1 Appendix) which contain open-ended questions and probes to explore the expectations and acceptability of a smart nursing home will be pilot tested. We plan to interview both the elderly people and the family members of eligible elderly people; however, due to the Covid-19 pandemic, it might not be feasible to travel to China to conduct face-to-face interviews; thus, we will prepare an alternative solution for the interviews, for example, by using voice call or voice message on WeChat. We will recruit 30 participants first, but the final sample size for this qualitative study will be determined when a point of data saturation is reached; for example, when additional data collection reveals no substantial new information [88]. Each in-depth interview will take between 30 and 45 minutes, and the focus group interview will take around 45 to 60 minutes. Prior to the interviews, participants will provide their sociodemographic and related information such as age, education, occupation, income, insurance, and health condition. Information about this research including the concept of smart nursing homes will be explained to all participants in Chinese before they are asked to provide an informed consent. All interviews including the interviews by using voice call on WeChat will be conducted in the Chinese language and audio-recorded. Investigators will take down field notes and transcripts will be translated line by line into English. For interviews conducted using voice messages on WeChat, the voice messages will be automatically converted into texts and translated into English on WeChat. All data will be translated into English to allow discussions with the non-Chinese speaking researchers.

**Qualitative data analysis plan.** Interviews will be video-recorded, and all data will be transcribed verbatim and managed using NVIVO 11 and ATLAS.ti8 software. Data in Chinese will be translated into English for co-investigators who are only proficient in English. Qualitative data will be analysed [89] to identify a thematic framework, guided by the smart technology adoption behaviours of the elderly consumers theoretical model [55]. The emerging codes will be analysed thematically according to the framework if they are coherent. The team members will repeatedly read each document and transcript to familiarise themselves with the whole data set and to ensure the accuracy of the transcription reviews. Codes will be applied to raw data, and then grouped into clusters to generate themes. Consensus to the codes and themes among at least three of the four investigators (ZYY, SJ, FKR, SSG and BHC) needs to be reached. The data will be compared in an ongoing manner to identify the true meaning of the data. Member checking and investigator's reflexivity will be used to increase the credibility and the reliability of the data. Data credibility will be enhanced through further analysis of interview and field notes, views from the different categories of elderly people of different sociodemographic backgrounds, for example, those from different regions, past occupation, and incomes as well as through triangulation involving multiple investigators. The content of the interviews will be shared with some participants, and their feedback on the findings will be sought for validation purposes (respondent validation). Investigators involved in the data analysis will engage in data collection and explain their biases or assumptions during data analysis and interpretation (reflexivity). Discussions of raw data, interpretations, and conclusions with peers (peer debriefing) will also be employed. Confirmability will be enhanced by keeping an audit trail of the audio/video files, electronic documents, original field notes, and translations [90].

## Phase II: Questionnaire development and validation

In Phase II, a questionnaire will be developed to assess the expectations and acceptability of a smart nursing home based on the qualitative data gathered in Phase I. The codes will become variables; themes will become scales; and the quotations from raw data will become questionnaire items. A 5-point Likert score will be used to measure the scales, and this will determine the level of expectations and acceptability. Based on the scores of the items or the summed scores of the items, expectations and acceptability can be categorized into 2 or 3 categories by combining the lower and upper responses into separate categories. This may be done at item level or at each of the construct level, depending on the dimensionality and psychometric property of the scale. The questionnaire will have three sections: 1) sociodemographic characteristics which include one-item on the willingness to move to a nursing home (Yes or No) [16]; 2) expectations of a smart nursing home; and 3) acceptability to adopt smart technologies in a nursing home. Clear instructions and descriptions of the definition, concept, and criteria of a smart nursing home will be provided at the beginning of the questionnaire. The smart technology adoption behaviours of elder consumers theoretical model [55] will guide the qualitative analysis and item design in the questionnaire. We expect the final version of the questionnaire to contain 30 items in Section 2 and Section 3 after conducting a member check and face and content validity in an expert committee. The expert committee comprises the investigators with expertise in qualitative research (SGS), statistical analysis (SJ), gerontechnology (FZR), and research methodology (BHC).

**Validation and reliability.** A cognitive debriefing involving 10 people to further assess the face and content validity of the questionnaire will be carried out [91]. Feedback will be discussed by the expert committee who will determine the appropriateness and comprehensiveness of the questionnaire, and the ethical considerations in the process. Face and content validity of the questionnaire will be assessed by using the COSMIN checklist (COnsensus-based Standards for the selection of health status Measurement INstruments) to ensure its relevancy, comprehensiveness and comprehensibility [92]. The COSMIN checklist was developed in an international Delphi study to evaluate the quality of selected measurement instruments. It measures internal consistency, reliability, measurement error, content validity (including face validity), structural validity, hypotheses testing, and cross-cultural validity (the three aspects of construct validity), criterion validity, and responsiveness [93]. Based on the checklist, items will be rated by experts in the team to establish face, content, and construct validity in Phase II. The duration to complete the questionnaire is estimated to be less than 30 minutes. Further validation of the questionnaire will be conducted in Phase III. Cronbach's alpha will be applied to prove internal consistency (0.70 or higher is considered as good) of the expectations and acceptability questionnaire [94]. Exploratory factor analysis using principal component analysis will be used to examine the structural validity of the expectations and acceptability questionnaire separately. In addition, the Kaiser-Mayer-Olkin test of sampling adequacy, Bartlett's test of sphericity, and the scree plot will be performed to check the unidimensionality of the expectations and acceptability questionnaire. The convergence criteria will be set at eigenvalue above 1. Construct validity (hypothesis-testing) will be tested when comparing the responses to the expectations or acceptability to the one-item on willingness to move to a nursing home (Yes or No) [16]. We hypothesize that the highest tertile expectation will be associated with willingness measured by at least an odd ratio of 2.0, and the highest tertile acceptability will be associated with willingness by at least an odd ratio of 3.0. It is also hypothesized that expectations and acceptability will correlate at a coefficient of at least 0.4. One-month intra-rater test-retest reliability will be carried out on a random sample of at least 50 elderly people [95] throughout the data collection period in Phase III.

## Phase III: A nationwide survey on expectations and acceptability of a smart nursing home among the mainland Chinese people

In Phase III, a nationwide survey by using the questionnaire that has been developed to examine the levels of expectations and acceptability of a smart nursing home among representative samples of mainland Chinese elderly people will be conducted. Additionally, we will also determine the sociodemographic factors associated with the different categories of expectations and acceptability of a smart nursing home among the mainland Chinese elderly people. Web-based applications such as the *wjx.cn* or *jinshuju.net* will be used to develop the questionnaire in Chinese. A written consent will be recorded as each participant clicks on the response button that indicates an agreement before the participant proceeds to answer the questionnaire and submit it online.

**Setting.**   In order to get the representative samples and potential target users of smart nursing homes from different parts of mainland China, four cities have been chosen: Xi'an, Nanjing, Shenyang, and Xiamen to represent the North, South, West and East of China, respectively. As one of the most popular tourist cities in Western China, Xi'an has 12.9 million population, and 19 national pilot projects on smart senior care have been conducted in the city [96]. As the capital city of Jiangsu Province in Eastern China, Nanjing has a high proportion of ageing population with about 30% of the city population being elderly people aged ≥65 years old [97]. Shenyang is one of the cities in Northern China which has a higher ageing population. The proportion of elderly people aged ≥65 years old is expected to be over 25% in the city by 2020 [98]. Xiamen is ranked as China's second most suitable city for living, and this has attracted many retired people to move to the city. The municipal government of Xiamen promotes smart healthcare to assist elderly people who live at homes independently, and it also encourages the provision of the services in institutional care for the elderly people [99]. These four cities are expected to have sufficient eligible samples of elderly people who could respond and complete the questionnaire.

**Participants.**   The inclusion criteria for recruiting representative samples are: 1) aged between 60 and 75 years old and live in mainland China; 2) being able to use WeChat App; and 3) being able to read. The exclusion criteria are: 1) elderly people who have already moved into a nursing home; 2) those with life expectancy predicted to be less than one year; and 3) those diagnosed with psychiatric disorders that may impair their ability to answer or complete the questionnaires. This does not exclude those who are treated and without cognitive impairment.

**Sample size estimation.**   Since there has been no previous similar study conducted, this study takes the approach of best estimation of the smallest difference in proportion of expectations and acceptability of a smart nursing home among the elderly people to be at 10% [16]. Using the G Power 3.1.2 with 0.90 power and significance at two-sided α of 0.05, the estimated sample size is 263. Based on our scoping review [75], the concept of a smart nursing home is relatively new, and exposure to this concept among the city-dwelling elderly people is believed to be more-or-less equal. Therefore, the estimation of sample size will not be adjusted for cluster design effect because individuals in the same city are believed to be no more alike to each other than they are to individuals in a different city in terms of expectations and acceptability of a smart nursing home. However, we will check the variances between the different cities in the analysis. Adaptively, if new evidence should appear that requires an adjustment on the estimated sample size for the cluster design effect, we will inflate by a factor of $1 + (n-1)\rho$, where n is the cluster size, and ρ is the intra-cluster correlation coefficient which we will obtain from the new study. Taking into consideration of about 50% non-response rate and incomplete questionnaires returned, the number of eligible respondents is at least 526. This sample size is

expected to be sufficient for a multiple logistic regression analysis with up to five determinants [16]. Having an adequate sample size will influence the quality and accuracy of the results including the psychometric testing of the questionnaire [100]. Generally, the guideline [101] suggests that 10 responses are needed per item, so a minimum of 100 participants will be required. The sample size estimation is also more than sufficient for an internal consistency and reliability testing among the samples. Based on our earlier experience and communication with experts in China, the required sample size of the eligible respondents through the online and in print questionnaires by post is feasible to be achieved. We will ensure the responses are unique without duplicates from the same person by requiring the participant's telephone number and address to where the token of participation will be sent to after the participant has submitted the completed questionnaire. The investigator (ZYY) will call the respondent to ensure that the participant has only answered once. An elderly couple in the same family will be considered as one participant.

**Measures.**   To answer study objectives 4–5, the questionnaire will consist of three sections: 1) sociodemographic characteristics; 2) willingness [16] to move to a nursing home and expectations on smart nursing home; and 3) acceptability in adopting smart technologies in a nursing home. The sociodemographic characteristics will include age, gender, marital status, education (no formal education, elementary school, middle school, high school and higher), occupation (employed, unemployed, retired), household monthly income per capita (<$375, $375–749, $≥750), insurance (UEBMI = Urban Employee Basic Medical Insurance; URBMI = Urban Resident Basic Medical Insurance; NCMS = New Cooperative Medical Scheme), living condition (with children or a spouse), number and type of chronic diseases and prior history of hospitalization. In Section 2 and Section 3 of the questionnaire, the 5-point Likert scales will be used to assess the main factors and items that have been developed in Phase I and to measure the level of expectations and acceptability of a smart nursing home, for example, expectations will be measured as: 1) much less than expected; 2) less than expected; 3) matched expectations; 4) exceeded expectations; and 5) greatly exceed expectations, and the acceptability of a smart nursing home will be measured as: 1) very unlikely; 2) unlikely; 3) not sure; 4) likely; and 5) very likely. A higher score will indicate a higher level of expectations and acceptability. The association of the sociodemographic factors with the different categories of expectation and acceptability will be determined through statistical analysis.

**Sampling process.**   The recruitment process of representative samples from Xi'an, Nanjing, Shenyang, and Xiamen will be carried out and is expected to be completed within two months. Firstly, eight contact persons, for example, two people each from Xi'an, Nanjing, Shenyang, and Xiamen will be appointed to help identify the local elderly WeChat groups and if necessary, to recruit the participants. Each contact person will provide five WeChat groups of the elderly communities in each city (a total of 10 elderly WeChat groups per city). Each WeChat group must have a minimum of 50 registered elderly people. Four to six WeChat groups from each city will be selected by using a web-based tool on *www.random.org.* The number of WeChat groups from each city is expected to provide a total of at least 132 (526/4) potential eligible elderly people. The remaining WeChat groups in each city will be kept as back-ups for future recruitment of participants. The link of an electronic flyer about this study, information on the token of appreciation, and online questionnaire (with a detailed information sheet and a consent form) will be disseminated through the selected WeChat groups. To ensure higher participation, reminder messages will be sent three times on a weekly basis. After two weeks, the same recruitment process by applying the same strategy in the remaining back-up WeChat groups from the city will be repeated if the required number of respondents has not been reached. Participants will be required to rate each item and complete all items in the questionnaire. Alternatively, for participants who are not likely to complete the

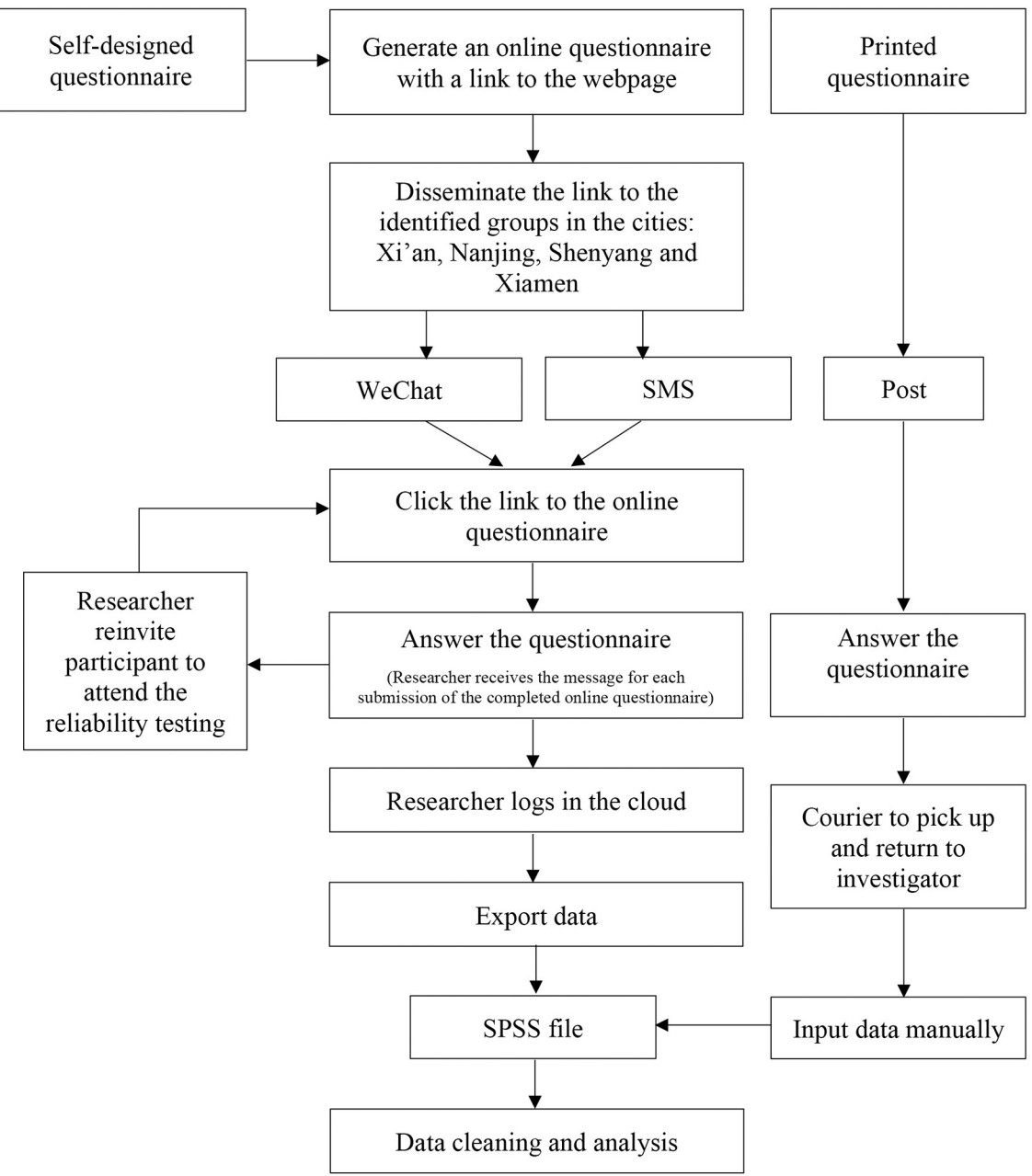

**Fig 3. Data collection process.**

questionnaire online, a hard copy of the questionnaire will be sent to them. Once they have completed the questionnaires, a courier service will pick up the questionnaires at no extra cost to the participants. Once there are at least 66 (263/4) completed questionnaires from each city, the principal investigator (ZYY) will release an announcement on the same elderly WeChat groups that the study has recruited the required sample size and will stop receiving further responses. The data collection process is shown in Fig 3. To recruit a random sample of at least 50 elderly people for the one-month intra-rater test-retest reliability testing, a total of 26 elderly people from each city will be re-invited (assuming the same 50% response rate) and given the

same explanations and instructions as well as the same payment amount of appreciation. These 26 elderly people will be identified from the respondents who have submitted their online questionnaires in the first month from each city. They will be evenly spread out over the four weeks (7 per week or one per day, the median respondent) if the total number of respondents over these four weeks is $\geq$ 26. Otherwise, we will apply the same principles of re-inviting the required number from each city, either over a shorter or longer period.

**Analysis plan.**   The data from the set of questionnaires will be exported to SPSS (Statistical Package for Social Sciences) for analysis. Questionnaires completed by post will be manually added to the data set by the investigators. Data cleaning will be done by checking that each data point is within plausible ranges or else, verification from the original data source will be conducted. We will also check the nature of missing data. If the missing data could be judged as at least missing at random and with a sufficient sample size, only the fully completed questionnaires will be used for analysis. If the case is otherwise, multiple imputation (with 10 runs) will be used to replace the missing data in the variables. Imputed variables will be set within a pre-defined possible range.

Descriptive statistics will be used to describe the characteristics of the respondents, and the levels of expectation and acceptability of a smart nursing home according to data distributions. Numerical variables will be expressed as means ± standard deviation or median and interquartile range, while categorical variables will be expressed as absolute frequencies. A chi-square test will be applied to examine the sociodemographic factors associated with the different categories of expectation and acceptability. With a 95% confidence interval, the *P*-value of less than 0.05 will be considered statistically significant.

The associated and independent factors from the sociodemographic and clinical variables on expectation and acceptability will be estimated in separate multiple logistic regression models. In the event of insufficient event rate and sample size for expectation and acceptability in categorical form, we may recode them into tertile and make the lowest score category as the referent group. Alternatively, multiple linear regression analyses with the two dependent variables as a continuous variable may be conducted. Any independent factor with a *P*-value < 0.20 from univariable regression analysis will be included in the multiple regression analysis. Multicollinearity between any independent variables will be checked according to the tolerance < 0.4 (Variance Inflation Factor $\geq$ 2.5). In the presence of multicollinearity, the more critical or essential variables from the clinical perspectives will be selected for use in the final regression analysis [102]. All models, Q-Q plots for normality, residual plots for linearity and homogeneity assumptions and model fitting will be checked.

Meanwhile, due to the lack of similar tools in construct validity and hypothesis testing, we will apply construct validity (hypothesis-testing) with one-item on willingness to move to a nursing home in the questionnaire to conduct the intra-rater test-retest reliability in Phase III.

## Discussion

This study may contribute to a better solution for the rapidly ageing society in China and may be relevant to a broader group of readers interested in this research or industrialised countries that provide nursing homes. It is believed that much innovation of smart technologies can be used and integrated to promote nursing home care within the field of health assessment, activities of daily living and care management that could improve the quality of life and quality of care in nursing homes [103–105]. Smart nursing homes will provide the elderly people an alternative solution with independent, safe and comfortable features to home-based care [47,48,106–109]. It could also be the solution to the existing nursing homes to adapt and transform to a tech-assisted nursing home operation and living environment that are welcomed by

the elderly people and their family members [55]. With a better understanding of the elderly people's expectations and acceptability of smart technologies in nursing homes, researchers, stakeholders, and policy makers may be able to effectively develop a complete smart nursing home model that incorporates appropriate technologies, integrates relevant medical services and yet remains affordable to benefit the majority of the elderly people [110]. This would facilitate a transformation of a traditional nursing home to a smart one including the necessary skillsets or required training for the nursing home operators and healthcare professionals to meet the expectations of all stakeholders [111].

To our best knowledge, this is the first research that explores an in-depth insight of expectations and acceptability of smart nursing homes among stakeholders, especially the mainland Chinese elderly people and their family members. The definition of a smart nursing home is first established through a scoping review. Through the mixed methods study approaches, we are able to gather the qualitative evidence on the concept of a smart nursing home, the expectations and acceptability of smart nursing homes among the Chinese elderly people and their family members. Conducting the interviews in Hainan and Dalian will capture a representative and variety of elderly Chinese from the different regions in China. Sequentially, a nationwide survey would assess the levels of the expectations and acceptability of a smart nursing home in a wider elderly population.

Due to the movement restriction during the Covid-19 pandemic, this study will utilize WeChat as the data collection tool. It is the most popular messaging application and the leader of mobile payment methods in China. At the bottom of the WeChat screen there are four icons: Chat, Contacts, Discover, and Me which support the functions of textual, pictorial, audio, video communication besides sharing contacts of others, self and creating groups of consenting individuals for a certain purpose [112]. WeChat covers a billion active users daily; thus, with WeChat, the research process can be conducted more effectively and efficiently. However, similar to other online surveys, this approach may face a relatively high non-response rate or incomplete responses to the questionnaire. Nevertheless, this could be minimized with a function from the online questionnaire platform that notifies the principal investigator when a submission is executed. The completeness of the questionnaire will be checked, and if the case is otherwise, the respondent will be informed and requested to complete the questionnaire before resubmitting it. All data underlying the findings will be fully available upon reasonable request on publication of the results.

We believe this research will be robust enough to achieve our intention to advance an evidence-based approach for a smart nursing home that is feasible, effective, and efficient for the ageing communities. We will compare and discuss the results of this study to other similar studies if they are conducted elsewhere, reported and published. Future research that takes into account the results of this study, cultures, and government policies will contribute to a better smart nursing home model for the rapidly ageing societies in China. The transformation of a smart nursing home model is potentially applicable to other developed and developing countries which consist of high proportion of ageing population, lack of medical resources and healthcare professionals, especially for the existing nursing homes with large number of residents or frail elderly people.

## Supporting information

**S1 Appendix. Semi-structured interview guides (for elderly people and the family members).**
(PDF)

## Author Contributions

**Conceptualization:** Yuanyuan Zhao, Boon-How Chew.

**Data curation:** Yuanyuan Zhao.

**Investigation:** Jing Su.

**Methodology:** Yuanyuan Zhao, Shariff-Ghazali Sazlina, Fakhrul Zaman Rokhani, Jing Su, Boon-How Chew.

**Project administration:** Yuanyuan Zhao.

**Supervision:** Yuanyuan Zhao, Shariff-Ghazali Sazlina, Fakhrul Zaman Rokhani, Boon-How Chew.

**Validation:** Shariff-Ghazali Sazlina, Jing Su, Boon-How Chew.

**Writing – review & editing:** Yuanyuan Zhao, Shariff-Ghazali Sazlina, Fakhrul Zaman Rokhani, Jing Su, Boon-How Chew.

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
