## [Decision Letter · Decision Letter 0]

14 Jan 2021

PONE-D-20-23914

Title: The Chinese elderly people’s expectation s and acceptability to a smart nursing home model: A mixed methods study protocol

PLOS ONE

Dear Dr. Chew,

Thank you for submitting your manuscript to PLOS ONE. After careful consideration, we feel that it has merit but does not fully meet PLOS ONE’s publication criteria as it currently stands. Therefore, we invite you to submit a revised version of the manuscript that addresses the points raised during the review process.

It is recommended to:

-Clarify the study motivation discussing the original contributions in comparison to existing works.

-Clarify the objectives of the study and make explicit the research questions and the research hypotheses.

-Revise and clarify theoretical framework section: for each mentioned construct provide definition, indicators, measurement tools, etc.

-Clarify how data collection tools are derived from research framework.

-Clarify the selection criteria of the two specific cities for the purpose of the interview.

-Clarify selection criteria of subjects for the semi-structured and the in-depth interviews.

-Include in the research  recent Chinese Smart Home care work.

-Extend the review section.

- Revise the abstract.

- Revise the English language.

-Improve the manuscript readability e. g. simplifying complex phrases.

For details see the enclosed reviewers' comments.

We look forward to receiving your revised manuscript.

Kind regards,

Filomena Papa

Academic Editor

PLOS ONE

Journal Requirements:

Reviewers' comments:

Reviewer's Responses to Questions

**Comments to the Author**

1. Does the manuscript provide a valid rationale for the proposed study, with clearly identified and justified research questions?

Reviewer #1: No

Reviewer #2: Yes

2. Is the protocol technically sound and planned in a manner that will lead to a meaningful outcome and allow testing the stated hypotheses?

Reviewer #1: Partly

Reviewer #2: Yes

3. Is the methodology feasible and described in sufficient detail to allow the work to be replicable?

Reviewer #1: No

Reviewer #2: Yes

4. Have the authors described where all data underlying the findings will be made available when the study is complete?

Reviewer #1: No

Reviewer #2: Yes

5. Is the manuscript presented in an intelligible fashion and written in standard English?

Reviewer #1: No

Reviewer #2: No

6. Review Comments to the Author

You may also provide optional suggestions and comments to authors that they might find helpful in planning their study.

Reviewer #1: The authors focus on a well-investigated topic in current literatures. I do not find how the proposed work is different from the existing ones that I have mentioned below in details. My decision for this manuscript will therefore be a reject.

1. The study motivation is not clear. The authors aim to investigate the elderly people’s intention towards using smart technologies in a nursing home setting. First, the term “smart technologies” must be clarified. It is too broad, and unless a proper scope of the technology is given it will be difficult to comprehend the usage scenario. Second, from the definition of nursing home it seems as if the authors are referring to some form of an old-age home. Generally nursing homes provide critical care too with doctors and nurses being present round-the-clock. Often, they have mini operation theaters and even an ICU/HDU unit in some cases. So, elaborate on this aspect that whether the acceptance model that you propose includes critical care or focus only on the general well-being aspect of the elderly. The Introduction section is too vague and the neither the research gaps nor the research contributions are clear. Moreover, all forms of healthcare institutes have already embraced various forms of ICT technologies for their day-to-day operations. Therefore, from an elderly perspective what aspects of smart technology do you plan to focus on is unclear as there are numerous works already available in this regard.

2. I do not get the point of presenting the section on theoretical framework. The factors that are proposed in Figure 1. are different from what is explained in this section. I do not find factors such as self-efficacy or openness of information in Figure 1. Moreover, how do you separate the expectation and the confirmation part of the theoretical model? Expectation is pre-usage phenomenon, while confirmation is the post-usage scenario. How do you handle this aspect in your model? There are no hypothesis and I am unsure what the authors are trying to claim. You must be having proper hypotheses for measuring the user perception towards smart technology, else I doubt what really is the objective and the outcome of this work. In short, what I mean is the theoretical framework section needs a major overhaul with proper hypotheses keeping in mind the research objectives.

3. In the study aims and objective section the authors state that “The aims of this study are to explore the expectations and acceptability of the mainland Chinese elderly people to smart nursing homes, and to determine the levels of these expectations and acceptability, as well as the sociodemographic factors associated with this matter”. The objective clearly points out that the authors are focusing on two distinct and theoretically different aspects of expectation and confirmation. So, please elaborate how to do so? Moreover, how do define and determine the different levels of expectation and acceptability? How many different levels do you have and how do you come up with these different levels? The information provided in this section is too vague and you must add more details. Second, smart nursing homes and smart homes are the same? If not, how they are different. I mentioned this point before also. How will be the perception of elderly people be different if focusing on smart nursing homes vs. smart homes? If you are trying to focus on various independent assistant living technologies (for otherwise healthy elderly people) for improving the quality of life, in that case I doubt the selection of nursing homes will not be a proper choice. So, please clarify on this aspect.

4. I am not sure about Objective #3. If you are basing this work on some already existing theoretical model (as shown in Figure 1), then why do you need to propose a new scale? In fact, the constructs that are presented in Figure 1 are not novel, and they have been used in numerous contexts over time. So, what is the purpose of developing a new scale, when you already have pre-existing ones? Similarly, objectives #4 and #5 lack clarity. As I mentioned before what do you mean by different levels? How do you differentiate between the different levels? What is your theoretical justification between the said different levels? There are just too many unknowns. Similarly, what are the different categories of expectation and how do you obtain them?

5. I am not sure of the rationale behind selecting the two specific cities for the purpose of the interview. What are the specialties of these cities and how can you ensure that these cities are representative of China in general?

6. Who will be the subjects for the semi-structured and the in-depth interviews? Are they going to be the same/different? Why/why not? How do you pre-determine the sample size to be around 50? What logic do you base this decision on? For example, the sample size has a high correlation to the number of questions that you are going to ask in the survey. Therefore, how do you select these numbers and ensure that it is optimum? The proposed interview durations are too long.

7. The proposed methodology and contributions are not novel in my opinion, and there are already a multitude of studies on this aspect.

Reviewer #2: Thank you for the research work and review report. The paper tries to focus on Smart Nursing home model with some recent work their advantages and disadvantages. There are few points that must be addressed

1. Paper does not have the meaningful and significant abstract.

2. Research does not include recent Chinese Smart Home care work.

3. The review is done very brief it does not cover the topic completely, it should be extended.

4. There are some lengthy sentences and complex phrases, these should be addressed to improve the readability.

7. PLOS authors have the option to publish the peer review history of their article (what does this mean?). If published, this will include your full peer review and any attached files.

Reviewer #1: No

Reviewer #2: **Yes: **HEMANT GHAYVAT

---

## [Author Response · Author response to Decision Letter 0]

17 Feb 2021

Dear Reviewers,

We would like to thank you for taking the time to review our manuscript and we thank you for your valuable comments. We have revised the manuscript accordingly. For response letter, please see our attached file 'Response to Reviewers'.

---

## [Decision Letter · Decision Letter 1]

4 May 2021

PONE-D-20-23914R1

Title: The expectations and acceptability of a smart nursing home model among Chinese elderly people: A mixed methods study protocol

PLOS ONE

Dear Dr. Chew,

Thank you for submitting your manuscript to PLOS ONE. After careful consideration, we feel that it has merit but does not fully meet PLOS ONE’s publication criteria as it currently stands. Therefore, we invite you to submit a revised version of the manuscript that addresses the points raised during the review process.

It is required to:

-Modify/integrate the introduction to clarify the gaps of knowledge in the field of acceptance of smart health care and the added value of the present research.

- Clarify the characteristics of target population considered in the study.

- Justify the selection of the adopted theoretical framework in comparison to the most relevant ones present in the literature.

- Provide more details in the explanation of figure 1 (theoretical framework).

- Expressly define the stage of technology adoption considered in the study.

- Clarify the kind of interviews developed in phase 1.

- Detail the procedures adopted respectively in focus groups, in-depth interviews and Wechat interviews.

It is also recommended to:

- Simplify long and complex sentences.

- Modify the first sentence of the introduction.

- Revise sections names when necessary.

- Fix some still existing English imperfections.

See  the enclosed comments for details.

We look forward to receiving your revised manuscript.

Kind regards,

Filomena Papa

Academic Editor

PLOS ONE

Additional Editor Comments (if provided):

Some clarifications are needed in the following sections.

Section "Theoretical framework"

Is the word “acceptance” used with the same meaning of “acceptability”? Please provide the definition of acceptability and also of the other constructs included in the model.

Section "sampling process"

-Clarify the kind of interviews developed in phase 1 i.e. focus group and in-depth interview.

How many people is involved in each focus group? Is in-depth interview individual?

-Detail the procedure adopted in focus groups, in-depth interviews and Wechat interviews

Specify if only members of the same family are included in a focus group.

Section data collection

Is “focus group” used with the same meaning of “semi-structured interviews”? Consider that a semi-structured interview could be also individual.

Reviewers' comments:

Reviewer's Responses to Questions

**Comments to the Author**

1. Does the manuscript provide a valid rationale for the proposed study, with clearly identified and justified research questions?

Reviewer #1: No

Reviewer #2: Yes

Reviewer #3: Partly

2. Is the protocol technically sound and planned in a manner that will lead to a meaningful outcome and allow testing the stated hypotheses?

Reviewer #1: Partly

Reviewer #2: Yes

Reviewer #3: Yes

3. Is the methodology feasible and described in sufficient detail to allow the work to be replicable?

Reviewer #1: No

Reviewer #2: Yes

Reviewer #3: Yes

4. Have the authors described where all data underlying the findings will be made available when the study is complete?

Reviewer #1: Yes

Reviewer #2: Yes

Reviewer #3: Yes

5. Is the manuscript presented in an intelligible fashion and written in standard English?

Reviewer #1: No

Reviewer #2: Yes

Reviewer #3: Yes

6. Review Comments to the Author

You may also provide optional suggestions and comments to authors that they might find helpful in planning their study.

Reviewer #1: I happened to review the first round of this paper. Unfortunately, I still have the following issues:

1) The Introduction is still vague. Acceptance of smart-healthcare by elderly people is a matured topic of research currently. I still cannot find a single paragraph which brings out the current drawbacks the existing literatures have. In fact, there are numerous literatures that propose some model using mixed methods/qualitative/quantitative methodologies for finding the perception of elderly people towards smart-healthcare. The current work is not novel, and how it tries to position itself apart from the current works is still not clear. Highlighting on the current drawbacks is extremely important for setting up the research context. Sadly, that is still missing, which creates doubt on the need, purpose, and novelty of the research.

2) The authors state that the elderly people considered in this study are “without terminal illnesses, not bed-bound or requiring critical care”. There is a concept of Quality of Life (QoL), which says that age is not a way by which a person will be classified as being an elderly or not, rather it’s the quality of life that will be the deciding factor. This is more true because of the smart technologies that are available now along with the advanced healthcare facilities that have substantially improved the existing QoL. The point I am trying to make is the basic purpose of smart technologies is promoting the concept of independent assisted living. AAL technologies can be implemented anywhere, even in a smart-home. This, together with the fact that the elderly people that are considered in this study are perfectly healthy ones, what is the need of selecting a nursing home setting? The one child policy and all other stuffs that are given as reasons by the authors do not hold true in a technology centric world as of 2021. In fact, one of the major focus of smart technologies is promoting independent living due to the small family sizes these days. Therefore, having the target population as perfectly healthy ones, and still focusing on nursing homes is difficult to digest for me, and the authors do not give any explanation for that.

3) While choosing a theoretical framework it is surprising that the authors do not consider any alternatives. There are several variations of technology adoption models related to senior people acceptance. Unless the most prominent ones are discussed, it is not convincing as to why a particular one should be used. I have not mentioned any specific healthcare related technology acceptance model for the seniors as there are a lot to choose from.

4) Objectives 1 and 2, are talking about expectations and confirmations. Please keep in mind that theoretically these are two different concepts and reflect two different stages of technology adoption. Expectations are prior to using any technology, and confirmations refer to a post adoption stage. Therefore, the study sample needs to be selected carefully and people should have experience in using a technology if the “confirmation” aspect is to be evaluated.

5) The purpose behind objective 3 is not clear. Considering there are so many models related to elderly technology adoption related to health what is need behind a new instrument? As I commented previously also the motivation behind the work is not properly set up, which questions the need of having yet another instrument in this already highly matured context.

6) In response to my previous comment #2, the authors have said that “The expectation and acceptability in the theoretical model are not two separate constructs but rather expectation is a determinant of acceptance if the smart nursing home appraisal meets the expectation”. I will refer to the paper on Expectation Confirmation Theory by Bhattacharjee et. al. in this respect. The way expectation and acceptability is presented in the current work, it is synonymous to expectation confirmation, which are two different constructs related to different stages of adoption. The term “acceptance”, itself means confirmation. Unless something is confirmed, it can never be accepted. The authors didn’t mention “intention to accept”, rather they said acceptance, which is nothing but confirmation. Moreover, if you simply replace acceptance with intention to accept, it does not make sense, because in that case how expectation is theoretically a determinant of intention to accept needs to be substantiated. The theory building has flaws and need to be examined at a greater depth.

7) My previous comment #3 is also unanswered. I had asked how do you define and determine the different levels/categories of expectation and acceptability? How many such different categories are there? These information are still not provided. What is the need for developing a new questionnaire? As I pointed out previously substantial literatures exists in the aspect of elderly healthcare. Therefore, first you need to identify the need of developing yet another questionnaire. Moreover, you are already using an existing framework, wherein the various factors are already pre-determined. This is not a Grounded Theory approach that based upon observations, a new theory is getting developed. In fact, that is not even the objective of the current work. What novel factors emerge out of the interview apart from the ones that are already discussed that necessitates the use of a questionnaire. If the Phase II is carried out for increasing the sample size only that’s a different story and not really a good way.

8) The study sample is not properly aligned to the study objectives. The sample of elderly people are perfectly healthy. The differentiation between the smart-homes and smart nursing-homes that the authors refer to can only be realized by including elderly people who are ill. Because the whole idea of “smart” is to promote AAL. Smart technologies are here to solve the problems the author’s mention. Therefore, a normal elderly people who has never been critically ill cannot have the right perception or differentiation between a smart-home and a smart nursing-home. The sample for me is therefore not a correct representation of what the authors are trying to claim. I mentioned this in my previous review too, but I find no explanation to this.

9) Coming back to the need of the new scale as I mentioned before there are many extant literatures that are focused on elderly in China. So, what is lacking in them that makes you propose a new one? Moreover, the authors state that “The new scale that is developed from the Phase I qualitative approach ensures it of high content and construct validity that may not be obtained by adopting scales from different cultures, settings and time.”. First, a high content and construct validity is already established in whatever scales a particular paper uses. Second, why not use existing scales that focus on a Chinese elderly context? You may modify them a bit to suite the present needs. Third, based on a cultural context how is China different from say non-Chinese countries? There are several literatures on elderly healthcare that focus on nearby countries like Taiwan, Japan, Thailand, Bangladesh, etc. Based on Hofstede’s 6D theory please explain the radical difference between China and these countries that I mentioned. As I said previously also, proposing a new scale is fine, as long as you have a reasoning to do so, which I am not finding for the present case.

Overall, I do not feel most of my previous comments have been answered. So, my decision for this version will still be the same as previous.

Reviewer #2: Thank you for the revision, it would be better if you simplify some of long and complex sentences. As the it will improve the readability of the article. One more thing this paper miss a meaningful block diagram which also needed.

Reviewer #3: I have been invited to review this paper in the second round.

The paper was greatly improved following thederive reviewers' comments and I believe it is coherent and deserves to be published.

I only have a couple of additional remarks.

The first sentence in the Introduction does not make much sense:"The term ‘nursing home’ comes from an ancient Greek word, ‘gerocomeia’." Although it is correct that the first nursing homes were the greek 'gerocomeia', the English term 'nursing home' obviously does not come from those. The sentence should be rephrased or joined to the second one.

There are two sections titled "Study aims and objectives" and "Study objectives". I think this is redundant. Please consider renaming the former as "Research questions".

Although I appreciate the polishing of the language, there are still many English imperfections that should be fixed.

7. PLOS authors have the option to publish the peer review history of their article (what does this mean?). If published, this will include your full peer review and any attached files.

Reviewer #1: No

Reviewer #2: **Yes: **Hemant Ghayvat

Reviewer #3: **Yes: **Bartolomeo Sapio

---

## [Author Response · Author response to Decision Letter 1]

23 Jun 2021

Dear Reviewers,

We would like to thank you for reviewing our manuscript again and we thank you for your comments. We have revised the manuscript accordingly. For the reply to your comments, please see our rebuttal letter in attached file. Thank you.

---

## [Decision Letter · Decision Letter 2]

16 Jul 2021

PONE-D-20-23914R2

Title: The expectations and acceptability of a smart nursing home model among Chinese elderly people: A mixed methods study protocol

PLOS ONE

Dear Dr. Chew,

Thank you for submitting your manuscript to PLOS ONE. After careful consideration, we feel that it has merit but does not fully meet PLOS ONE’s publication criteria as it currently stands. Therefore, we invite you to submit a revised version of the manuscript that addresses the points raised during the review process.

It is requested to revise the manuscript according to Reviewer2's comments (see after).

We look forward to receiving your revised manuscript.

Kind regards,

Filomena Papa

Academic Editor

PLOS ONE

Journal Requirements:

Reviewers' comments:

Reviewer's Responses to Questions

**Comments to the Author**

1. Does the manuscript provide a valid rationale for the proposed study, with clearly identified and justified research questions?

Reviewer #2: Yes

Reviewer #3: Yes

2. Is the protocol technically sound and planned in a manner that will lead to a meaningful outcome and allow testing the stated hypotheses?

Reviewer #2: Yes

Reviewer #3: Yes

3. Is the methodology feasible and described in sufficient detail to allow the work to be replicable?

Reviewer #2: Yes

Reviewer #3: Yes

4. Have the authors described where all data underlying the findings will be made available when the study is complete?

Reviewer #2: Yes

Reviewer #3: Yes

5. Is the manuscript presented in an intelligible fashion and written in standard English?

Reviewer #2: Yes

Reviewer #3: Yes

6. Review Comments to the Author

You may also provide optional suggestions and comments to authors that they might find helpful in planning their study.

Reviewer #2: Thank you for the research on elderly care and nursing. Please address the comments below:

1. The research need to describe in detail how these protocol could solve the low adaptability of elderly nursing care

2. Please make a comparative analysis with other available elderly care protocol with the proposed one.

3. Describe the controlled environment context in detail where it could be implemented

4. Revise the abstract and conclusion to improve the readibility.

Reviewer #3: All comments have been addressed. The paper can now be published.

7. PLOS authors have the option to publish the peer review history of their article (what does this mean?). If published, this will include your full peer review and any attached files.

Reviewer #2: **Yes: **hemant ghayvat

Reviewer #3: **Yes: **Bartolomeo Sapio

---

## [Author Response · Author response to Decision Letter 2]

21 Jul 2021

Dear Reviewers,

We would like to thank you for reviewing our manuscript again and we thank you for your comments. We have revised the manuscript accordingly. For the reply to your comments, please see our rebuttal letter in attached file. Thank you.

---

## [Decision Letter · Decision Letter 3]

27 Jul 2021

Title: The expectations and acceptability of a smart nursing home model among Chinese elderly people: A mixed methods study protocol

PONE-D-20-23914R3

Dear Dr. Chew,

We’re pleased to inform you that your manuscript has been judged scientifically suitable for publication and will be formally accepted for publication once it meets all outstanding technical requirements.

Kind regards,

Filomena Papa

Academic Editor

PLOS ONE

Additional Editor Comments (optional):

Reviewers' comments:

Reviewer's Responses to Questions

**Comments to the Author**

1. Does the manuscript provide a valid rationale for the proposed study, with clearly identified and justified research questions?

Reviewer #3: Yes

2. Is the protocol technically sound and planned in a manner that will lead to a meaningful outcome and allow testing the stated hypotheses?

Reviewer #3: Yes

3. Is the methodology feasible and described in sufficient detail to allow the work to be replicable?

Reviewer #3: Yes

4. Have the authors described where all data underlying the findings will be made available when the study is complete?

Reviewer #3: Yes

5. Is the manuscript presented in an intelligible fashion and written in standard English?

Reviewer #3: Yes

6. Review Comments to the Author

You may also provide optional suggestions and comments to authors that they might find helpful in planning their study.

Reviewer #3: All comments have been addressed.

The paper can now be published.

7. PLOS authors have the option to publish the peer review history of their article (what does this mean?). If published, this will include your full peer review and any attached files.

Reviewer #3: **Yes: **Bartolomeo Sapio

---

## [Editor Report · Acceptance letter]

11 Aug 2021

PONE-D-20-23914R3 

The expectations and acceptability of a smart nursing home model among Chinese elderly people: A mixed methods study protocol 

Dear Dr. Chew:

I'm pleased to inform you that your manuscript has been deemed suitable for publication in PLOS ONE. Congratulations! Your manuscript is now with our production department. 

Kind regards, 

on behalf of

Dr. Filomena Papa 

Academic Editor

PLOS ONE